# Effects of Electrical Stimulation on Delayed Onset Muscle Soreness (DOMS): Evidences from Laboratory and In-Field Studies

**DOI:** 10.3390/jfmk8040146

**Published:** 2023-10-13

**Authors:** Maristella Gussoni, Sarah Moretti, Alessandra Vezzoli, Valerio Genitoni, Guido Giardini, Costantino Balestra, Gerardo Bosco, Lorenza Pratali, Elisabetta Spagnolo, Michela Montorsi, Simona Mrakic-Sposta

**Affiliations:** 1Institute of Chemical Sciences and Technologies “G. Natta”, National Research Council (SCITEC-CNR), 20133 Milan, Italy; maristella.gussoni@unimi.it; 2National Research Council (IFC-CNR), 20159 Roma, Italy; morettisarah099@gmail.com; 3Institute of Clinical Physiology, National Research Council (IFC-CNR), 20159 Milan, Italy; alessandra.vezzoli@cnr.it (A.V.); lorenza.pratali@cnr.it (L.P.); elisabetta.spagnolo@cnr.it (E.S.); 4Centro Neo-Medico Qi Gong People Milan, 20133 Milan, Italy; valerio.genitoni@gmail.com; 5Neurology and Neurophysiology Department, Mountain Medicine Center Valle d’ Aosta Regional Hospital Umberto Parini, 11100 Aosta, Italy; ggiardini@ausl.vda.it; 6Società Italiana Medicina di Montagna, SIMeM, 35138 Padova, Italy; 7Motor Sciences Department, Physical Activity Teaching Unit, Université Libre de Bruxelles (ULB), 1050 Brussels, Belgium; cbalestra@he2b.be; 8Environmental Physiology & Medicine Lab, Department of Biomedical Sciences, University of Padova, 35131 Padova, Italy; gerardo.bosco@unipd.it; 9Department of Human Sciences and Promotion of the Quality of Life, San Raffaele Roma Open University, 00166 Roma, Italy

**Keywords:** exercise, pain, oxy-inflammation, fatigue, micro-invasive, ultra-runners, non-contracting electrical stimulation

## Abstract

Intense, long exercise can increase oxidative stress, leading to higher levels of inflammatory mediators and muscle damage. At the same time, fatigue has been suggested as one of the factors giving rise to delayed-onset muscle soreness (DOMS). The aim of this study was to investigate the efficacy of a specific electrical stimulation (ES) treatment (without elicited muscular contraction) on two different scenarios: in the laboratory on eleven healthy volunteers (56.45 ± 4.87 years) after upper limbs eccentric exercise (Study 1) and in the field on fourteen ultra-endurance athletes (age 47.4 ± 10.2 year) after an ultra-running race (134 km, altitude difference of 10,970 m+) by lower exercising limbs (Study 2). Subjects were randomly assigned to two experimental tasks in cross-over: Active or Sham ES treatments. The ES efficacy was assessed by monitoring the oxy-inflammation status: Reactive Oxygen Species production, total antioxidant capacity, IL-6 cytokine levels, and lactate with micro-invasive measurements (capillary blood, urine) and scales for fatigue and recovery assessments. No significant differences (*p* > 0.05) were found in the time course of recovery and/or pre–post-race between Sham and Active groups in both study conditions. A subjective positive role of sham stimulation (VAS scores for muscle pain assessment) was reported. In conclusion, the effectiveness of ES in treating DOMS and its effects on muscle recovery remain still unclear.

## 1. Introduction

As widely reported [1,2], regular exercise is beneficial to health, while unaccustomed or exhaustive exercise can produce detrimental effects such as muscle damage, inflammation, and oxidative stress, especially when eccentric contractions are involved [3]. Damage to muscle fibers leads to the leakage of intracellular enzymes, such as creatine kinase and lactate dehydrogenase [4,5], as well as to the increase in perceived muscle soreness, which is often referred to as delayed-onset muscle soreness (DOMS). DOMS occurs 8–24 h after strenuous exercise, its peak rising 24–48 h after [6]. Characterized by dull and aching pain, DOMS is usually felt when exercised muscles are moved, stretched, or palpated and is often accompanied by increased tenderness and stiffness [7,8,9].

Exercise-induced Reactive Oxygen Species (ROS) production has been suggested as a contributory mechanism to muscle damage [10]. Continuously body-generated, ROS are promptly inactivated by antioxidant defenses [11]. Required for normal force production, a low ROS amount modulates cell signaling processes, while a higher ROS concentration reduces force production in both a time-and dose-dependent way, thus contributing to muscle fatigue [12]. Furthermore, following a muscle-damaging exercise, an acute inflammatory response takes place with an increase in circulating immune cells [13]. This response acts to mediate the process of necrotic tissue breakdown, whereby macrophages’ inflammatory-related cytokines release [14] and a consequent increase in c-reactive protein (crP) concentration occurs [15,16]. Despite the large variability in such responses [17], taken together, these observations can be considered acute markers of exercise-induced muscle damage, providing an indication for the subsequent recovery [18,19]. Amateur and professional athletes are concerned about muscular discomfort and pain phenomena because they can limit further exercise and training activity [20]. Many researchers have investigated various treatment strategies aimed at alleviating DOMS symptoms, restoring the maximal muscle function as rapidly as possible and/or reducing the magnitude of the initial injury [21,22].

Recent studies have found that the most effective recovery methods are those improving blood circulation, fluid filtration or reabsorption in the microcirculation, which are capable of both blood lactate concentration and muscle soreness decrease [23,24,25]. Delayed muscle soreness onset may also be found after an endurance exercise; so, ultra-endurance athletes are prone to DOMS. In this regard, some methods to relieve DOMS were studied in trained endurance athletes in order to reduce recovery times [26,27]. DOMS treatment strategies have included nutritional [28] and pharmacological interventions [29], massage [30], low-intensity exercise [31], ultrasound application [32], and cryotherapy [33]. The potential benefits purported by these interventions are those of an altered hemodynamics to facilitate a greater removal of tissue damaging molecules and a reduction in localized edema formation [34].

Long before man had discovered the scientific principles behind electricity, since the times of the ancient Egyptian kingdom, passing through the Roman emperor Tiberius and arriving at John Walsh, the mystery of electricity in the biological/medical field has always commanded great fascination [35] To date, many devices used microcurrents in the sports and/or medical fields in Europe, Asia, and the Americas. Therefore, electro-stimulation devices are adopted for both acupuncture and transcutaneous electrical nerve stimulation (TENS) [36,37,38]. The devices deliver electrical impulses through electrodes placed on the skin surface near nerves or at trigger points [39]. Classical acupuncture points are proximate to peripheral nerves and named meridians.

Particularly, neuromuscular electrical stimulation techniques can activate the skeletal muscle pump via the transcutaneous stimulation of muscle fibers, in turn increasing local blood flow. Therefore, even if with conflicting results, a great number of studies have investigated the potential of reducing muscle soreness when adopting different electrical stimulation techniques and methods [40,41], while the electrical stimulation (ES) efficacy in DOMS prevention and/or treatment and its effects on muscle recovery remain still unclear [41].

The present study aims to investigate, by micro-invasive methods, the effect of non-contracting ES [42] on both the sensation of muscle soreness and muscle strength development. The research has been conducted on two different scenarios: in the laboratory, on healthy volunteers after concentric–eccentric exercise of upper limbs (biceps brachii, Study 1), and through an in-field study on ultra-endurance athletes after an ultra-running race, measuring the lower limbs (quadriceps, Study 2).

## 2. Materials and Methods

All experiments were carried out testing the effects of electrical stimulation (ES) treatment on the sensation of muscle soreness and muscle recovery. All recruited participants received an explanation of the study purposes, risks, and benefits before the experiment, after which they read and signed a specific informed consent form. The study consisted of two parts: in the laboratory (on biceps brachii, Study 1) and in the field (on quadriceps, Study 2). The method herein presented adopted reliable, simple, and micro-invasive measurements to test the ES efficacy by monitoring the oxy-inflammation status throughout the assessment of different parameters: some of them were determined in both 1 and 2, others were determined only in 2.

### 2.1. Physiological Parameters, Scales for Physical Fatigue and Recovery Assessments (Study 1 and Study 2)

The physiological parameters were determined by a bipolar bio-impedentiometry (TBF-300A Body Composition Analyzer; Tanita Corporation, Arlington Heights, IL, USA); finger O_2_ saturation (SaO_2_) and Heart Rate (HR) (Oximetry—Ohmeda TuffSat—GE Healthcare^®^, Helsinki, Finland) were measured in both Study 1 and 2, while blood pressure (BP), measured by a standard cuff sphygmomanometer (OMRON M7 Intelli IT, Omron, Japan), was recorded only in Study 2. The subject-perceived exertion and the muscle fatigue were assessed by means of the Borg Rate of Perceived Exertion scale (RPE) [43]. The Visual Analogue Scale (VAS) was adopted, too. This latter is a simple, efficient, and reliable method widely employed both in research and clinical practices [44,45,46,47]. The method returns a measure of subjective mood (happy/unhappy), general wellness (rested/tired), general sensation (hot/cold, calm/anxiety, headache, nausea), and muscle-pain intensity. The scale consists of a 100 mm line where “0” indicates the absence of pain, while “100” indicates the worst ever felt pain. Each subject had to locate the position on her/his body where the pain was felt. The quality of race recovery was assessed at the end of the race and post-ES treatments (T1 and T2), comparing to pre-race (T0), by the Total Quality of Recovery scale (TQR) proposed by Kenttä and Hassmén [48].

### 2.2. Blood Measurements (Study 1 and 2)

For each recruited subject, capillary blood (500 μL) was taken from the fingertip: ROS production rate, antioxidant capacity, lactate, and IL-6 concentration were determined. Plasma samples were obtained by the centrifugation (Thermo Scientific™ Medifuge Small Benchtop Centrifuge, Monza, Italy) of a heparinized microvette (Sarstedt^®^, Nümbrecht, Germany). All samples were initially stored at −80 °C in multiple aliquots and thawed only once before analysis.

***Reactive Oxygen Species (ROS)*** measurement was performed by an Electron Paramagnetic Resonance (EPR) X-band spectrometer (E-Scan-Bruker^®^ BioSpin, GmbH, Billerica, MA, USA) at 37 °C. The ROS production rate was calculated from the EPR spectra on blood samples treated with CMH (1-hydroxy-3-methoxycarbonyl-2,2,5,5-tetramethylpyrrolidine) solution (1:1). [49,50,51,52]. All spectra were collected by adopting the same acquisition parameters and handled by the standardly supplied by Bruker^®^ software (Win EPR System, V. 2.11). All data were, in turn, converted in absolute concentration levels (μmol.min^−1^) by adopting CP• (3-carboxy-2,2,5,5-tetramethyl-1-pyrrolidinyloxy) stable radical as an external reference.

***Antioxidant capacity (TAC)*** was measured from blood samples (10 μL), using a commercial EDEL potentiostat electrochemical analyzer (Edel Therapeutics^®^, Switzerland) equipped with a redox sensor in a three-electrode arrangement, following a previously described method [52,53,54]. The blood sample was loaded onto a chip and a potential, increasing from 0 to 1.2 V, at a scan rate of 100 mV.s^−1^ (versus Ag/AgCl reference electrode), was applied, while the resulting current was measured at a carbon electrode (WE). Data were expressed in nW.

***Blood lactate concentration (La_[b]_)*** was obtained from blood (0.2 μL) by fingertip (Lactate scout; EKF^®^, Italia, Milano, Italy).

***Interleukin-6 (IL-6)*** levels were determined on plasma by an ultra-sensitive ELISA kit (R&D Systems^®^, Minneapolis, MN, USA), according to the manufacturer’s instructions by a previously described method [52,55,56,57]. The signal was spectrophotometrically (Infinite M200, Tecam, Austria) measured at a wavelength of 450 nm.

### 2.3. Urine Measurements (Study 2)

Urine samples were collected by the participants of the field study (2), voluntary voiding in a sterile container before (T0) and after ES treatments (T1). Urine samples were stored in multiple aliquots at −80 °C until assayed. Creatinine and neopterin concentrations were measured by an isocratic high-pressure liquid chromatography (HPLC) method, as previously described [52,55,56,57].

### 2.4. Thermographic Imaging (Study 1)

Skin surface temperature measurements were performed by using a FLIR 320A Infrared thermocamera (FLIR Systems^®^, Wilsonville, OR, USA) with 320 × 240 pixels image resolution, a 7.5 to 13 mm spectral range, a detector time constant (about 12 ms, 62% reading) and <0.07 °C at 30 °C sensitivity. The camera was connected to a personal computer, using the ThermaCAM Researcher Software. The subject was placed in the room 1 m from the camera. Room temperature (~22 °C) and relative humidity (~50%) were detected by a thermos-hygrometer (Oregon Scientific^®^, Portland, OR, USA). Once the parameters, such as emissivity, were provided, the images were stored for computer processing. The first thermogram was taken at rest before starting the exercise session, the second was taken at the end of the exercise, and the third was taken after the ES treatments. Using the Flir Tools^®^ 4.1 software (FLIR Systems^®^, Wilsonville, OR, USA), the regions of interest (ROIs) were delimited in the thermogram by selecting the corresponding areas of the biceps brachii (between the axillary line and the elbow) and the temperature data returned by the Software.

### 2.5. Isometric Handgrip Strength Test (Study 1)

In the laboratory Study 1, the upper limb strength was estimated by the hand grip, measured from the exercised arm, by means of an electronic hand dynamometer (Deyard^®^ EH 101, Grandado, Germany). To measure the strength, the subject was asked to sustain his maximal strength for 5 s while receiving verbal encouragement. The highest value (less than 10% difference from three tests) was used for analysis. Before performing the test, each subject was asked to indicate the dominant upper limb: all reported being right-handed [58].

### 2.6. Low-Intensity Electro-Stimulation (ES) (Study 1 and 2)

All subjects underwent ES measurements (Active condition). The adopted device (MYOGEN^®^ WeCare Technologies, Milano, Italy) certificate CE, in security class IIA (CE n. 0068/QCO-DM/050-2018, Medical device registered with the Italian Ministry of Health BD/RDM 1748158), was a low-intensity electro-stimulator (ES) in the 1–100 Hz frequency range by a complex waveform, 1 to 500 µA output current, nominal tension 110–230 V/50–60 Hz, and absorbed power 25 W, which was automatically selected by a real-time analysis resident computer software. After skin cleaning, the electrodes were applied to biceps brachii (Study 1) or quadriceps (Study 2) using an electrolytes interface solution. The stimulation adopted in our study was as follows: monolateral in the laboratory study (Study 1); bilateral in the field study (Study 2). The device stimulated at a 0–100 Hz range frequency, by a 500 μA complex waveform, which was auto-determined by the instrument without eliciting any muscular contraction. A Sham stimulation (participants did not receive any current) was adopted as a placebo-controlled condition.

### 2.7. Laboratory Study—1, Participants and Experimental Protocol

Eleven healthy volunteers (4 females/7 males; 56.45 ± 4.87 years) without present or previous upper arm injuries and not suffering from any arm pain were recruited. A medical questionnaire was completed before participation in the study. Subjects were requested not to change their lifestyle and dietary habits and not to take any anti-inflammatory drug or nutritional supplements during the experimental period. All tests were performed under close medical supervision. DOMS was induced in the elbow flexors through repeated eccentric/concentric muscle contractions. All subjects performed, in a sitting position, repetitive flexion/extension movements of the elbow with a 3 kg handheld load [10]. Subjects were asked to use a smooth controlled movement: the time from the full flexion to the full extension was 3 s. This pattern was repeated until exhaustion. Voluntary exhaustion was defined as the inability to continue the elbow flexion/extension despite vigorous encouragement by the operators as well as by maximal levels of self-perceived exertion using VAS and RPE scales [43,44,45]. Each participant was asked to mark the level of muscle pain while his elbow joint was at 90° flexion: that is, when the muscle was in isometric contraction [59]. Participants were assigned to complete two experimental conditions in cross-over: physical exercise, followed, for a time not exceeding 10 min, by (a) Sham and (b) Active ES treatment. Each experimental condition was followed by a washout period of 15 days.

The two conditions were applied randomly to the participants who did not know which condition was applied. In fact, the supplied electric current was not perceptible, so there was not any recognizable difference between Active or Sham stimulations. The experimental protocol is displayed in the sketch of Figure 1. All variables were measured at rest (T0), immediately at the end of the exercise (T1), post-ES 10 min (T2) and 1 h after (T3). Muscle soreness was also assessed 24 h after exercise (T4) by the adopted RPE and VAS Scales.

All procedures were conducted according to the Declaration of Helsinki, and approval was obtained from the institutional Ethics Committee of the Aosta Hospital (n.895; 31 August 2015), Italy.

### 2.8. Field Study—2, Participants and Snapshot of the Race

The field Study 2 was performed during the 1st edition of the trail running XL “Tot Dret” (TD), which was organized in Aosta Valley (Italy), with a 134 km distance and 10,970 m+ elevation gain. Running along the route of the Alta Via number one, the Tot Dret is a short journey amid Mont Blanc (4810 m), Monte Rosa (4634 m) and Cervino (4478 m). The starting point was in Gressoney St Jean, the ending was in Courmayeur, and the maximum race time was 44 h.

Twenty-five experienced ultra-marathon runners (2 females/23 males; age 47.4 ± 10.2 year; height 175.1 ± 5.4 cm) voluntarily participated in the study. During the pre-race session (T0), a questionnaire was administered to collect data on the participants’ training experience. On average, the participants reported 5.3 ± 2.7 years of ultra-endurance experience. Training sessions in preparation for the race consisted of 4–5 sessions per week (about 59 ± 4.5 km/wk). No specific limitations were imposed regarding the use of vitamin/minerals supplements, herbs and medications; furthermore, a complete assessment of each participant’s health status (e.g., hypertension, allergies) was performed. Measurements (physiological values, and ROS production by blood) were performed all together at rest (T0) and immediately after ES (T1), while physical fatigue and recovery scale assessments were also performed 24 h post-race (T2). In Figure 2, the elevation profile of the race and the experimental protocol are shown.

### 2.9. Statistical Analysis

Statistical analysis was performed using the GraphPad Prism package for Mac (GraphPad Prism 9.5.1, Software Inc., San Diego, CA, USA) and statistical package IBM SPSS Statistics (IBM SPSS Statistics v. 22, Armonk, NY, USA) software. The data were expressed as mean ± standard deviation (SD). The normality of the data distribution was tested with the Shapiro–Wilk test. To test the assumption of homogeneity in variance, Levene’s test of equality of error variances was applied. Experimental data were compared using ANOVA repeated measures with a Dunn’s multiple comparison post hoc test; the *p* < 0.05 statistical significance level was accepted. Eta squared (η^2^) was calculated for estimates of effect size: small (0.0099 < η^2^ ≤ 0.0599), moderate (0.0599 < η^2^ ≤ 0.1399), and large (η^2^ ≥ 0.14) effects, respectively. The prospective calculation of the power to determine significant numbers was made by using the Freeware G*Power 3.1.9.6 software (http://www.psycho.uni-duesseldorf.de/abteilungen/aap/gpower3/, 2 September 2023) choosing an ROS production variable as the primary outcome [45]. At a power of 80%, the calculated number of significant subjects was 11, which was sufficient for the laboratory study (1) and below the subject’s population recruited for the field study (2).

## 3. Results

### 3.1. Laboratory Experiment—Study 1

#### 3.1.1. Physiological Parameters

Age, Height, Body Mass, BMI, Fat Mass, Free Fat Mass, and Total Body Water parameters were collected from all subjects and are reported in Table 1. Oxygen saturation (SaO_2_ %) and heart rate data (beats per minute; BPM) measured at rest (T0), at the end (T1), and 10 min after the two experimental session modalities (T2, Sham/Active) are displayed in Table 2. Data did not differ significantly.

#### 3.1.2. Laboratory Study 1, Biological Parameters (ROS, TAC, Lactate, IL-6)

The time course of the ROS production rate levels (mean ± SD) during the two test sessions (Sham/Active) is displayed in Figure 3A. Starting from not significantly different basal levels (T0), the ROS production rate significantly increased at the END of the exercise (T1, *p* < 0.001–0.0001) in both the Sham (Black symbols/lines) and Active (red symbols/lines) (Sham: 2.21 ± 0.16 vs. 2.49 ± 0.15 μmol.min^−1^; Active 2.21 ± 0.16 vs. 2.47 ± 0.19 μmol.min^−1^) test groups. After Sham or Active ES treatments (post-ES, T2), ROS levels returned almost at the basal values but still showed a low significant difference (*p* < 0.01).

At the same time, the antioxidant capacity levels (TAC), shown in Figure 3B, significantly decreased at the end of the exercise (*p* < 0.001–0.0001); (Sham: 117.3 ± 23.4 vs. 155.5 ± 35.4 nW; Active 120.7 ± 27.8 vs. 147.7 ± 31.4 nW) returning to the basal levels both after Sham or Active ES treatments (post-ES, T3).

Blood lactate concentration (La_[b]_) levels (see Figure 3C), while not significantly different at rest, significantly increased in both the two sessions (*p* < 0.0001) at the end of the exercise (Sham: 2.47 ± 0.57 vs. 1.25 ± 0.35 mM; Active 2.37 ± 0.47 vs. 1.27 ± 0.25 mM). Thereafter, lactate concentration came back toward the resting levels both after ES Sham or Active (10 min) treatments (post-ES, T2) as well.

As can be observed in Figure 3D, a significant increase (*p* < 0.0001) of the inflammatory state was suggested in both experimental sessions by the IL-6 levels evaluated at the end exercise (T1, Sham vs. Active: 2.04 ± 0.19 and 2.19 ± 0.28 pg·mL^−1^) and 1 h after the ES (Sham vs. Active: 1.99 ± 0.21 and 2.15 ± 0.25 pg·mL^−1^) with respect to rest (T0, Sham vs. Active: 1.53 ± 0.20 and 1.59 ± 0.22 pg·mL^−1^).

No significant differences (*p* > 0.05) were assessed between Sham and Active groups for all the variables assessed.

The results of Levene’s test and of the effect size (partial eta squared, ηp^2^) tests on the biological parameters analyzed in the laboratory study (1) are provided in Table 3. All variables met the required homogeneity by Levene’s test, and the effect size of Active versus Sham stimulations for ROS, Il-6, TAC, Lactate, and T° variables were calculated.

#### 3.1.3. Isometric Handgrip Strength and Scores (RPE, VAS)

In Figure 4A, the RPE score at the end of the exercise (T1) for the athletes assigned to the Sham or Active groups is reported. Most athletes rated the exercise effort as “somewhat hard/hard”. No significant difference was found between the two groups.

In Figure 4B, the VAS scores for subjective muscle pain assessment are reported. Significant (*p* < 0.05) differences among evaluation times were found in the Sham and Active groups. The VAS score was significantly increased in both sessions (*p* < 0.001) at T1 of the exercise compared to T0 (Sham: 52.7 ± 7.9; Active 54.5 ± 10.4). A significant difference (*p* < 0.001) at 1 h post-ES (T3) between Sham and Active groups was also observed (Sham vs. Active: 31.8 ± 7.5 vs. 19.1 ± 10.4, respectively), persisting also 48 h after the end of the exercise (T4: 41.9 ± 12.5 vs. 10.9 ± 10.4, respectively).

Figure 4C shows the maximal isometric handgrip strength (N) for each time-point of the study design. No significant differences (*p* > 0.05) were found between the two groups (Sham vs. Active). However, it is interesting to note how in the Active group, an increase in strength restoration at T2 (T2 vs. T1, Sham: 289.7 ± 85.47 vs. 309.0 ± 78.99 N; Active: 320.2 ± 106.6 vs. 313.2 ± 97.6 N) and 1 h post-exercise (T3) (T3 vs. T1, Sham: 301.7 ± 89.3 vs. 309.0 ± 78.99 N; Active 342.8 ± 107.0 vs. 313.2 ± 97.6 N) can be observed.

#### 3.1.4. Skin Temperature

Skin temperature data and examples of Infrared Thermographic Images (IRTs), with color-coded temperature maps, are shown in Figure 5 for the Sham and Active groups. All images were normalized to the same high reference level. No statistically significant changes in skin temperature were found between the Sham (T0: 30.84 ± 1.43 °C; T1: 33.75 ± 1.56 °C) and Active (T0: 30.47 ± 1.06 °C; T1: 34.44 ± 1.16 °C) groups, suggesting, as expected, that the exercise played the same effect on the two groups. A significant change (*p* < 0.0001) at T2 was found in Active study sessions with respect to Sham (Sham vs. Active; 31.81 ± 1.78 °C vs. 29.42 ± 1.86 °C: −6% vs. −14.5%). The different temperature change is also well shown by the color-coded images in Figure 5C and Figure 5C’.

### 3.2. Field Experiment, “Tot Dret”—Study 2

#### 3.2.1. Physiological Parameters

Eleven athletes (age 45.82 ± 8.51 year; body mass 73.23 ± 10.25; BMI 23.52 ± 2.76) dropped out of the race. Fourteen athletes (67%) completed the race in a mean time of 35.23 ± 4.52 h and were randomly divided into two groups (seven athletes per group). One group was treated by active ES, while the other one was treated by Sham. All athletes’ anthropometric and physiological parameters are reported in Table 4. Significant differences are reported.

#### 3.2.2. Biological Parameters

The ultrarace induced a wide increase in the biomarkers measured by capillary blood. As can be observed in the histograms of Figure 6, starting from not significantly different basal levels (*p* > 0.05) between the two athletes’ groups (Active vs. Sham), the ROS production rate (**A**: *p* < 0.05 Active: 1.72 ± 0.26 vs. 2.23 ± 0.24 μmol.min^−1^; Sham 1.67 ± 0.29 vs. 2.21 ± 0.31 μmol.min^−1^), lactate levels (**E**; *p* < 0.05–0.01; Active: 1.26 ± 0.4 vs. 3.63 ± 0.75 mM; Sham 1.41 ± 0.41 vs. 4.23 ± 0.91 mM), and IL-6 concentration (**F**; *p* < 0.05–0.01; Active: 1.56 ± 0.59 vs. 17.84 ± 9.06 pg.mL^−1^; Sham 1.41 ± 0.63 vs. 20.68 ± 9.04 pg.mL^−1^) significantly increased either after Active or Sham ES at post-race (T1).

At the same time, the antioxidant capacity levels (TAC), shown in Figure 6B, significantly decreased either after Active or Sham ES at post-race (*p* < 0.05–0.01; Active: 136.60 ± 12.62 vs. 108.0 ± 19.37 nW; Sham 138.4 ± 15.94 vs. 108.7 ± 14.73 mW). No effect was reported on total GSH and Hb (Figure 6C and D, respectively).

Moreover, after Active or Sham ES treatment at post-race (T1), a significant increase (*p* < 0.05–0.01) in creatinine (A; Active 2.1 ± 0.5 vs. Sham 2.1 ± 0.7) and neopterin concentrations (B; Active 106.9 ± 26.0 vs. Sham 90.38 ± 19.8) from urine samples was observed as shown by the histograms of Figure 7, starting from not significantly different basal levels (T0, *p* > 0.05) between the two athletes’ groups (ES Active vs. ES Sham).

The results of Levene’s test and of the effect size (η^2^) tests on the biological parameters on Active versus Sham stimulation analyzed in the field experiment—Tot Dret (Study 2) are provided in Table 5. As already reported in the laboratory Study 1 (Table 3), all variables met the required homogeneity by Levene’s test.

#### 3.2.3. Scores (RPE, VAS, TQR)

In Figure 8A, the RPE score after the ES treatment at post-race (T1) is reported. Most athletes rated the effort made during the exercise as “hard/very hard” (Active score 17.9 ± 1.4; Sham score: 18.4 ± 0.9).

Figure 8B shows VAS scores for subjective muscle pain assessment at post-race (T1) and at 24 h post-race (T2). Significant differences at T1 vs. T2 were found in both Active and Sham groups. In particular, the VAS score significantly increased in both sessions (*p* < 0.01) at the end of ES treatment (T1) post-race compared to T0 (Active 70.0 ± 11.2 vs. 2.4 ± 3.6; Sham: 69.6 ± 13.7 vs. 2.3 ± 3.7) and post-ES 24 h (T4, Active 63.0 ± 12.2; Sham: 66.1 ± 13.6). No differences were found in Active vs. Sham groups at T1 and T2 (*p* > 0.05).

In Figure 8C, the TQR scores are displayed: significant decreases were recorded at T1 (Active 6.28 ± 0.48 vs. Sham: 6.29 ± 0.49) and T2 (Active 8.57 ± 1.51 vs. Sham: 8.43 ± 1.71) in both ES groups vs. T0 (Active 20.0 ± 0.0 vs. Sham: 20.0 ± 0.0).

## 4. Discussion

Intense exercise, training and competition can induce tissue vibrations [60] and eccentric contractions [61], possibly leading to muscle damage [4], with a consequent temporary reduction in muscular force [62,63] and decreased physical performance [63,64]. At the same time, an increased tissue inflammation [60,61,62,63,64,65,66], perceived fatigue [61], pain [62,63], delayed onset of muscle soreness (DOMS) [64], and/or increased risk of injury [66,67,68] may occur.

As already pointed out, many literature studies investigated the potential of muscle soreness reduction by adopting different electrical stimulation devices but returning conflicting results [40]. This study’s results also appear conflicting.

The present study was carried out adopting a mini-invasive analytic technique capable of contemporarily investigating the changes in the level of several oxy-inflammation markers. In the light of them, the main finding of this study was that ES treatment does not objectively reduce DOMS. As a matter of fact, the two approaches presented, in the laboratory (1) and in the field (2), show data that do not always agree: this is linked to the type of exercise (anaerobic intense eccentric exercise/aerobic prolonged resistance exercise, respectively). However, post-ES treatments, in the examined cases, no adverse events were reported.

As previously reported [50,52,69], exercise induces changes in oxidative stress and inflammation: an increase in ROS production, lactate concentration, and IL-6, with a TAC decrease in both experimental conditions 1 and 2 found in the present study as well. In particular, the sharp rise of oxy-inflammation biomarkers (ROS and IL-6) levels, following the exercise, was a typical indicator of the inflammatory response [52,69].

In accordance with Dupuy et al., [70] the present study data did not show a significant difference between ES Sham and ES Active as concerns capillary blood ROS production and plasma IL-6 concentration responses when following both anaerobic as well as aerobic prolonged resistance exercise. In addition, the second study suffers the impact of environment hypoxia on the analyzed biomarkers and parameters, overall ROS levels and all parameters related to the metabolic and oxidative responses. This was the main reason why the results obtained by the two studies were not compared in this study. The effect of hypoxia just on ultra-endurance racers was previously studied by some of us [47,71].

Moreover, in a field study (2), a systemic oxy-inflammation could also be associated to the altered biochemical parameters measured in urine and in particular to the increased neopterin concentration, as previously reported [52].

Similarly, the variations (T0 vs. T1) reported in Figure 6, Figure 7 and Figure 8 were due to the race. Between the values recorded at T1 after the ES (both Sham and Active), no significant difference was observed. Again, in Figure 8, no difference between the values recorded at T1 (post-ES Sham or Active) and at T2 (1 h after ES Sham or Active) was observed.

The increase/decrease in the assessed biomarkers reported in Figure 3 at T1 were the same for both experimental sessions, and they were due to the exercise performed. After the ES (both Sham and Active), no significant differences were observed between the two treatments. Only a significant difference in VAS score after 1 h persisting also at 48 h post-ES between Sham and Active groups was observed.

The precise mechanism underpinning the soreness sensation is still not fully understood; most likely, it involves multifactorial processes, each presiding at different time-points throughout the period following eccentric exercise.

Initial nociceptors stimulation by products of tissue breakdown and/or edema is thought to occur within the first day of damage. In the present study, the collected thermographic images suggested an ES treatment effect at this early phase ascribable to an analgesic effect of the afferent nerve fibers’ stimulation as well as edema reduction possibly due to the increased microvascular blood flow [39].

Cheng and colleagues have reported that the application of a micro-Amp current to isolated rat skin is able to alter membrane transport, protein synthesis and ATP generation [71]. Indeed, subjects affected by DOMS may certainly present alterations in muscle membrane transport/trafficking, protein synthesis and degradation and ATP generation: the so-hypothesized effects could be therefore useful to restore cellular homeostasis [72]. Despite the controversial results found in the literature, the most relevant effect found in the present study was a best lowering of the temperature, measured by thermography in the post-exercise experimental condition, after Active ES, with respect to Sham ES.

In synthesis, when considering the results coming from this double study, showing that ES treatment has no effect on the measured biomarkers, the subjective positive role derived from ES stimulation must be anyway emphasized. In fact, a better effect on the fatigue sensation of decreased muscle pain after active ES was found in the laboratory study (1) 1 h after (ES Active 64% vs. ES Sham 38%) and up to 24 h after (ES Active 79% vs. ES Sham 21% respect END exercise) [70]. These findings were also confirmed in the field study (2) by VAS post-24 h ES (ES Active 21% vs. ES Sham 4.5%), even if this was valid for both ES Active and Sham treatments. Finally, the TQR scale showed a subjective post-competition recovery in both stimulation modalities (Active ES and Sham) as well, suggesting the placebo effects of the ES treatment [73].

Summarizing, the results of the present study did not show significant differences between ES and Sham control groups in reducing DOMS. Nevertheless, the placebo (or the sham stimulation) effect can play an important role in that athletes and competitors feel better recovered, more willing and able to perform subsequent training sessions or repeated tournament games.

## 5. Practical Applications and Strengths of the Study

The practical applications and strengths of the present study follow:

The biological measurements have been conducted by adopting micro-invasive methods.The ES treatments were conducted in two different types of exercise, one involving the upper limbs and the other involving the lower ones.

Future research/studies will consider team sports (i.e., volleyball, soccer) where all athletes show almost the same level of training/performance/fitness. Moreover, it could be possibly applied to a post-injury status to verify recovery times and/or to injury prevention. Finally, it could indicate prevention and treatment for the overtraining syndrome.

## 6. Conclusions

The data of the present study showed that the non-contracting ES treatment after aerobic and anaerobic intense exercise did not modify the values of the measured biological parameters. Rather, it played a positive perceptual benefit on fatigue sensation, which was possibly linked to a more rapid reduction in body surface temperature after exercise. These findings lead us to conclude that ES does not treat DOMS as well, as it does not help in promoting muscle recovery immediately as well as 24 h after the intervention. The effects played 48, 72, 96 h after the intervention and a higher number of subjects recruited could be considered a possible matter of further research.

## Figures and Tables

**Figure 1 jfmk-08-00146-f001:**
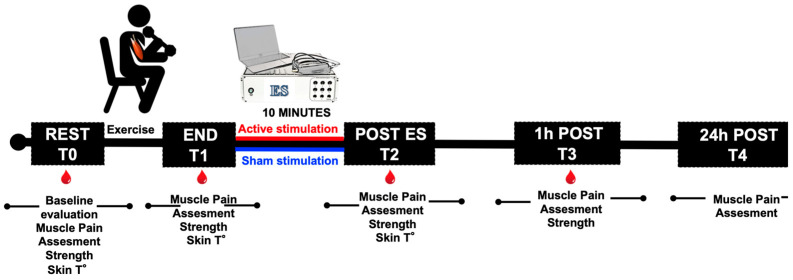
Laboratory Study (1). Experimental protocol design. As indicated, all subjects were tested at rest (T0), after exercise (T1), after electrical stimulation (ES, T2), and 1 h and 24 h post-exercise (T3 and T4). Active stimulation (red line) or Sham (blue line) were applied for 10 min at the end of the exercise. The red drops indicate the timing of the blood samples collection.

**Figure 2 jfmk-08-00146-f002:**
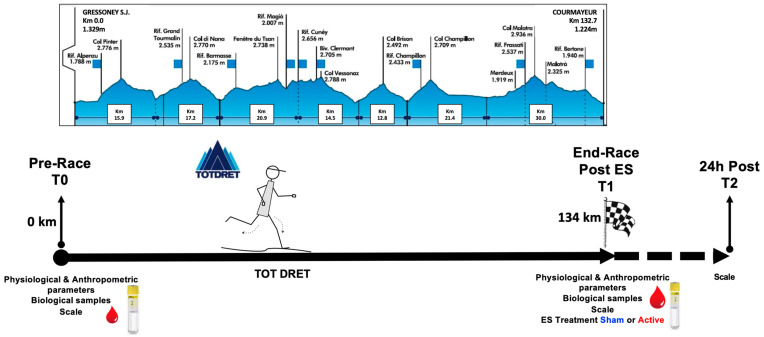
Elevation profile of the race (Tot Dret, TD) (http://www.tordesgeants.it (16 September 2015)) and experimental protocol adopted to measure biomarkers on the biological samples collected from the selected participants.

**Figure 3 jfmk-08-00146-f003:**
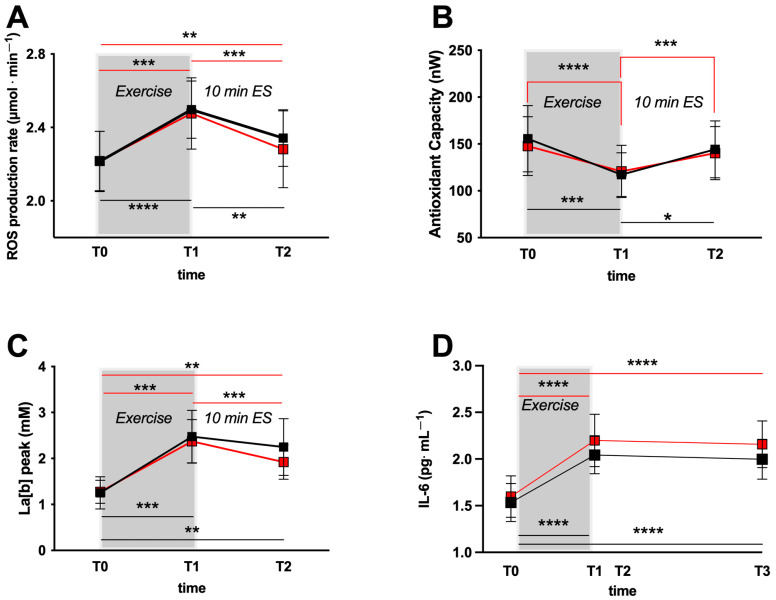
Time course of: (**A**) Reactive Oxygen Species (ROS) production rate (μmol.min^−1^) calculated from the Electron Paramagnetic Resonance (EPR) spectra, (**B**) total antioxidant capacity (TAC; nW), (**C**) blood lactate concentration (La_[b]_; mM), before (rest, T0), immediately after the exercise (end, T1) and after the electrical stimulation (ES = 10 min duration; T2). (**D**) Time course of interleukin-6 (IL-6; pg/mL) concentration at rest, at the end of the exercise and 1 h post-electrical stimulation (ES = 1 h; T3). Data were obtained during the two experimental sessions: Sham (black squares), Active (red squares) and expressed as mean ± SD. Lines are drawn as eye guide. * *p* < 0.05, ** *p* < 0.01, *** *p* < 0.001, **** *p* < 0.0001 symbols show statistically significant differences.

**Figure 4 jfmk-08-00146-f004:**
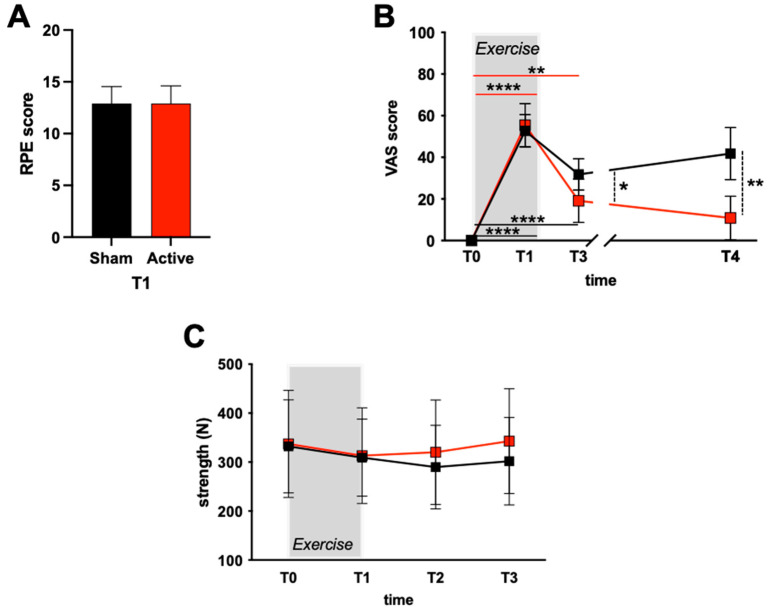
(**A**) Borg scale at end exercise (T1); (**B**) time course of the VAS score; (**C**) maximal isometric handgrip strength (N). Data were obtained from the two groups: Sham (black squares), Active (red squares) and expressed as mean ± SD. Lines are drawn as eye guide. Statistically significant differences symbols: * *p* < 0.05; ** *p* < 0.01; **** *p* < 0.0001.

**Figure 5 jfmk-08-00146-f005:**
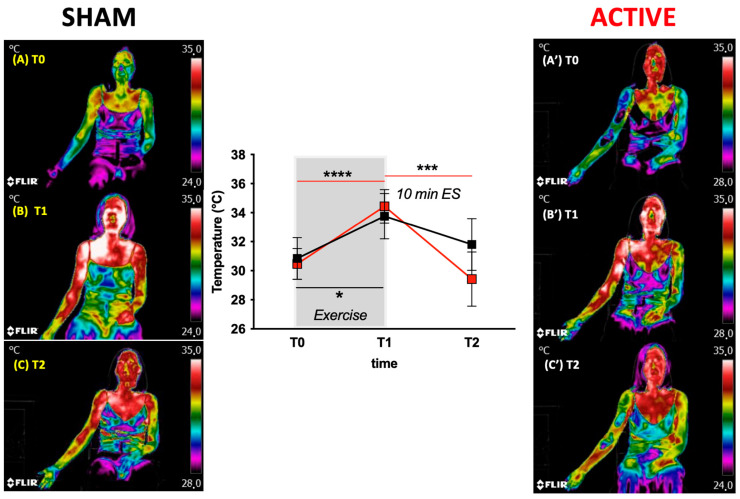
Examples of thermographic images with color-coded temperature maps; in the middle of the figure, the skin temperature (C°) data recorded at rest (T0), at the end of the exercise (end, T1) and after the electrical stimulation (T2, post-ES 10 min duration) for Sham (black squares–lines) and Active (red squares–lines) groups. Data are expressed mean ± SD. Statistically significant differences symbols: * *p* < 0.05; *** *p* < 0.001; **** *p* < 0.0001. At the left and right sides of the panel, anterior body infrared thermal images at T0 (**A**,**A’**), T1 (**B**,**B’**) and T2 (**C**,**C’**). All images were normalized to the same high reference level.

**Figure 6 jfmk-08-00146-f006:**
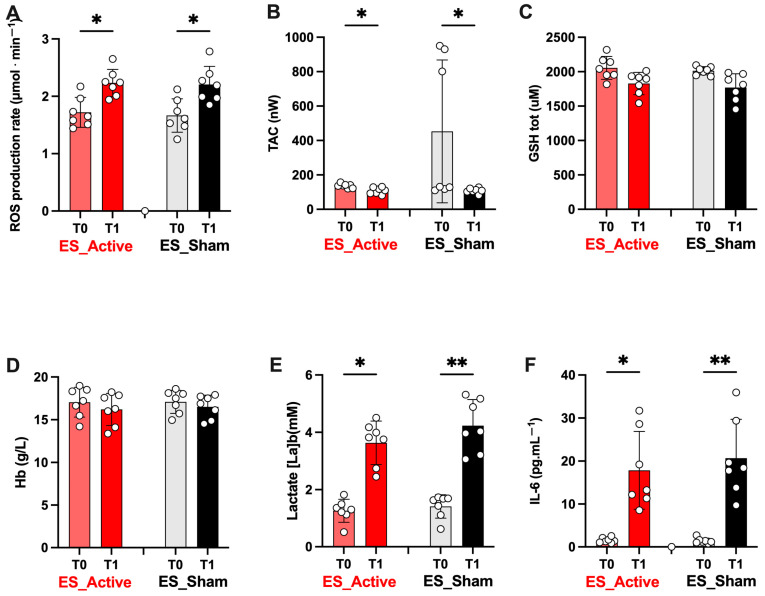
The electrical stimulation (ES) treatment effect after ultrarace (T1) on oxidative stress (OxS) is shown by the histogram plots of (**A**) Reactive Oxygen Species (ROS) production rate (μmol.min^−1^), (**B**) Total Antioxidant Capacity (TAC; nW), (**C**) total glutathione (tot GSH; μM), (**D**) hemoglobin (Hb; g/L), (**E**) blood lactate concentration ([La]b; mM), and interleukin -6 (IL-6; pg · mL^−1^) (**F**). Light red and light gray bars: ES_Active and ES_Sham treatments at T0; ES_Active (red bars) and ES_Sham treatments (black bars) at T1. On each bar, single data are displayed by empty circles. Data are expressed as mean ± SD. Statistically significant differences symbols: * *p* < 0.05, and ** *p* < 0.01.

**Figure 7 jfmk-08-00146-f007:**
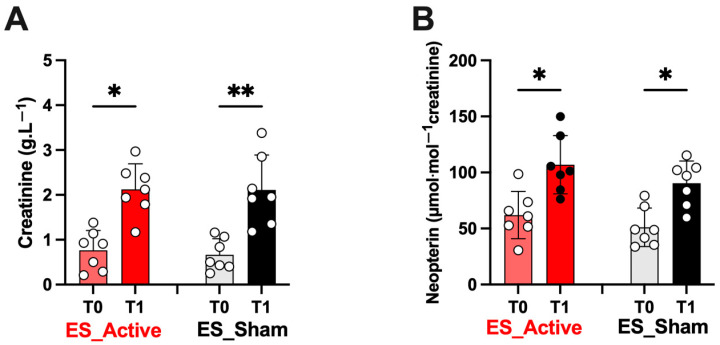
Electrical stimulation (ES) treatment effect after ultrarace (T1) on (**A**) creatinine (g · L^−1^) and (**B**) neopterin (μmol/min^−1^ creatinine) obtained from urine samples in both groups. Results are expressed as mean ± SD. Light red and light gray bars: ES_Active and ES_Sham treatments at T0; ES_Active (red bars) and ES_Sham treatments (black bars) at T1. On each bar, single data are displayed by empty circles. Data are expressed as mean ± SD. Statistically significant differences symbols: * *p* < 0.05, and ** *p* < 0.01.

**Figure 8 jfmk-08-00146-f008:**
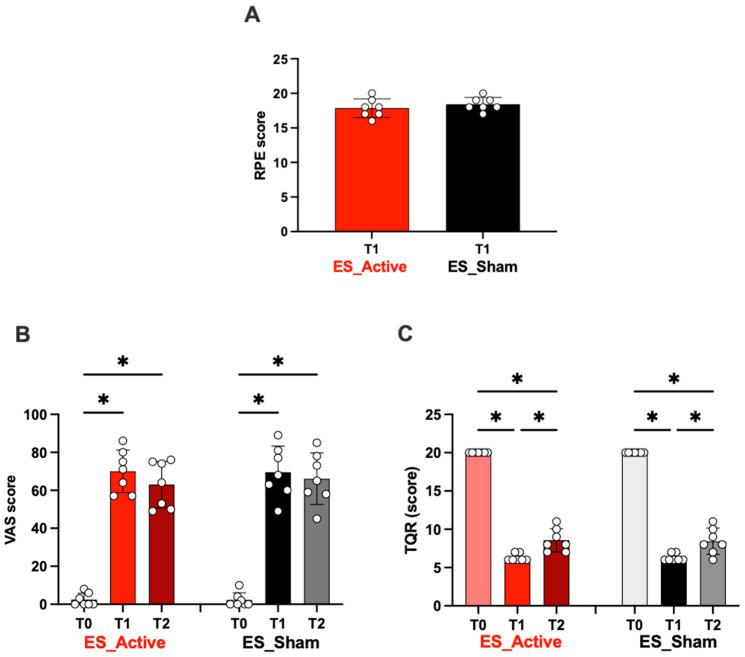
The electrical stimulation (ES) treatment effects after (T1) ultrarace on (**A**) Borg scale; (**B**) VAS and (**C**) TQR scores. Data are expressed mean ± SD. In light red and light gray, ES_Active and ES_Sham treatments at T0; in red ES_Active and black ES_Sham treatments at T1. Statistically significant differences symbols: * *p* < 0.05.

**Table 1 jfmk-08-00146-t001:** Laboratory study (1): physiological parameters collected from all subjects. BMI: body mass index; TBW: total body water. Data are expressed as mean values (±SD).

Physiological Parameters	
Age (years)	56.07 ± 4.87
Height (m)	1.71 ± 0.09
Gender	6 Females/7 Males
Body Mass (kg)	73.98 ± 15.54
BMI (kg∗m^−2^)	25.22 ± 4.11
Fat Mass (kg)	19.77 ± 7.10
Free Fat Mass (kg)	54.21 ± 12.86
TBW (kg)	37.62 ± 9.13

**Table 2 jfmk-08-00146-t002:** Laboratory study (1): oxygen saturation (SaO_2_ %) and heart rate (BPM) parameters collected from all subjects at different times of Active or Sham stimulation sessions (Rest, T0; End, T1; and post-ES, T2). Data are expressed as mean values (±SD).

	Sham Stimulation			Active Stimulation		
	T0	T1	T2	T0	T1	T2
SaO_2_ (%)	97.5 ± 0.7	96.8 ± 1.1	97.4 ± 1.0	97.4 ± 0.8	96.3 ± 1.4	97.2 ± 0.9
HR (bpm)	71.3 ± 8.1	85.7 ± 20.7	76.4 ± 10.9	71.00 ± 7.7	86.9 ± 19.7	77.0 ± 9.4

**Table 3 jfmk-08-00146-t003:** Test of between-subjects effect by groups: Levene’s test of equality of error variances and effect size estimates (η^2^) of the treatments on ROS, Il-6, TAC, lactate, and T° variables.

	Levene’s Test—EXP Lab (1) Active vs. Sham Stimulation			Effect Size- EXP Lab (1)
	T0	T1	T2	
ROS	F = 0.02929 *p*-value = 0.8649	F = 0.28806 *p*-value = 0.5943	F = 0.37283 *p*-value = 0.5447	η^2^ = 0.36
IL-6	F = 2.10077 *p*-value = 0.1546	F = 0.49079 *p*-value = 0.4907	F = 0.20208 *p*-value = 0.2020	η^2^ = 0.68
TAC	F = 0.00375 *p*-value = 0.9518	F = 0.2018 *p*-value = 0.6581	F = 0.04002 *p*-value = 0.8434	η^2^ = 0.19
Lactate	F = 0.40928 *p*-value = 0.5258	F = 1.07138 *p*-value = 0.3065	F = 0.82274 *p*-value = 0.3695	η^2^ = 0.56
T°	F = 0.63232 *p*-value = 0.4313	F = 2.7096 *p*-value = 0.1079	F = 4.24296 *p*-value = 0.4596	η^2^ = 0.53

**Table 4 jfmk-08-00146-t004:** All athletes’ anthropometric and physiological parameters. Mean (±SD) of the parameters collected from all athletes at pre-race (T0); all finishers at pre (T0) and after ES treatments at post-race (T1), and finishers divided into the ES Active and ES Sham session groups. BMI: Body Mass Index; FM: Fat Mass (kg); FFM: Free Fat Mass (kg); TBW: Total Body Water (kg); SaO_2_: arterial Oxygen Saturation (%); HR: Heart Rate (bpm); SBP: Systolic Blood Pressure (mmHg); DBP: Diastolic Blood Pressure (mmHg), and average time of completing the race (hours). Significant differences (T1 vs. T0) between all finishers as well as ES Active vs. ES Sham groups are reported: * *p* < 0.05; ** *p* < 0.01, *** *p* < 0.001 symbols.

Athletes’ Groups Tot Dret							
	ALL Athletes at (n = 25)	Finisher Athletes (n = 14)		ES—Active Group (n = 7)		ES—Sham Group (n = 7)	
	Pre-Race T0	Pre-Race T0	Post-Race T1	Pre-Race T0	Post-Race T1	Pre-Race T0	Post-Race T1
**Age (years)**	47.4 ± 10.2	48.0 ± 10.7	-	47.0 ± 12.1	-	49.0 ± 10.0	-
**Body Mass (kg)**	70.6 ± 10.1	70.0 ± 10.4	67.1 ± 9.5 ***	70.7 ± 8.7	68.7 ± 8.7 *	69.3 ± 12.5	65.5 ± 10.63 *
**BMI(kg·m^−2^)**	23.0 ± 2.5	22.5 ± 2.4	22.0 ± 2.4 ***	23.0 ± 2.2	22.3 ± 2.4 *	22.0 ± 2.6	21.6 ± 2.5 *
**FM (kg)**	7.9 ± 4.0	7.4 ± 4.3	4.9 ± 2.2 ***	7.8 ± 4.0	6.1 ± 2.5 *	8.7 ± 4.7	4.7 ± 2.2 *
**FFM (kg)**	63.6 ± 8.5	62.7 ± 8.7	60.7 ± 8.3 **	64.7 ± 6.3	62.3 ± 6.1 *	60.7 ± 10.7	58.4 ± 9.9 *
**TBW (kg)**	41.5 ± 5.6	41.6 ± 5.3	40.6 ± 5.0 *	41.9 ± 3.7	41.3 ± 6.9	41.3 ± 4.0	39.9 ± 6.1
**SaO_2_ (%)**	97.5 ± 1.0	97.7 ± 0.8	96.1 ± 0.6 ***	97.6 ± 0.9	96.0 ± 0.8 *	97.8 ± 1.0	96.2 ± 0.5 *
**HR (bpm)**	63.5 ± 11.2	59.1 ± 7.9	84.8 ± 7.9 ***	58.7 ± 8.2	82.4 ± 6.9 *	59.5 ± 8.4	87.1 ± 7.5 *
**SBP (mmHg)**	128.6 ± 14.7	125.9 ± 17.5	141.9 ± 14.3 ***	125.0 ± 7.6	141.7 ± 10.8 *	126.7 ± 24.49	142.0 ± 18.1 *
**DBP (mmHg)**	76.1 ± 9.5	73.6 ± 8.8	82.6 ± 5.7 ***	75.7 ± 8.8	84.9 ± 5.8 *	71.6 ± 9.0	88.6 ± 5.3 *
**Race completion time (h)**	32.15 ± 5.18	-	35.45 ± 4.98	-	35.31 ± 5.57		35.59 ± 4.75

**Table 5 jfmk-08-00146-t005:** Test of between-subjects effects by group: Levene’s test of equality of error variances and effect size (η^2^) of the treatments on ROS, IL-6, TAC, GSH, lactate and Hb variables.

	Levene’s Test—EXP in Field (B) Active vs. Sham Stimulation		Size Effect—EXP in Field (B)
	REST—T0	END—T1	
**ROS**	F = 0.16619 *p*-value = 0.6868	F = 3.25446 *p*-value = 0.0828	η^2^ = 0.50
**IL-6**	F = 0.01206 *p*-value = 0.9133	F = 1.36679 *p*-value = 0.2529	η^2^ = 0.73
**TAC**	F = 1.17291 *p*-value = 0.3000	F = 1.33197 *p*-value = 0.2709	η^2^= 0.48
**GSH**	F = 0.48966 *p*-value = 0.4902	F = 0.20224 *p*-value = 0.6566	η^2^ =0.41
**Lactate**	F = 4.06207 *p*-value = 0.0543	F = 2.31256 *p*-value = 0.1404	η^2^ = 0.74
**Hb**	F = 0.19891 *p*-value = 0.6592	F = 6.42359 *p*-value = 0.0176	η^2^ = 0.06
**Creatinine**	F = 1.09116 *p*-value = 0.3058	F = 0.02641 *p*-value = 0.8721	η^2^ = 0.67
**Neopterin**	F = 1.29507 *p*-value = 0.2659	F = 0.99735 *p*-value = 0.3275	η^2^ = 0.53

## Data Availability

All data are contained within the article.

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
