# Peer review of "Effects of Electrical Stimulation on Delayed Onset Muscle Soreness (DOMS): Evidences from Laboratory and In-Field Studies"

_jfmk, 2023, doi:10.3390/jfmk8040146_

Round 1

Reviewer 1 Report (New Reviewer)

In this elegant cross-over study the authors aimed to evaluate the effect of micro electrical stimulation (ES) in reducing/contrast the delayed onset muscle soreness (DOMS) after upper limbs eccentric exercise (lab test in healthy volunteers) or after ultra-endurance race in a group of athletes. The efficacy of a pre-set ES protocol was evaluated by measuring the oxy-inflammation status in blood and urine and by questionnaire scales for fatigue and recovery assessments. As the authors conclude, the main message of this study is that there were not significant effects elicited by ES in treating DOMS, with exception of a subjective positive role, derived from VAS scores for muscle pain assessment.

The manuscript is interesting, well written and fluent, and scientifically sounds.

 General comments:

Line 189-190: The authors say: The device stimulated at a 0-100 Hz range frequency, by a 500μA complex waveform, auto-determined by the instrument without eliciting any muscular contraction. The authors should provide a more detailed description of the ES protocol, indicating the pulse duration, and the frequency applied. Also is not clear if the stimulation was a bilateral stimulation, and voltage applied.

The second study was a high altitude ultra-endurance race, would be interesting include in discussion a consideration on possible differences observed across the two studies in view of a possible impact of environment hypoxia on the analysed biomarkers and parameters.

Author Response

Comments and Suggestions for Authors

In this elegant cross-over study the authors aimed to evaluate the effect of micro electrical stimulation (ES) in reducing/contrast the delayed onset muscle soreness (DOMS) after upper limbs eccentric exercise (lab test in healthy volunteers) or after ultra-endurance race in a group of athletes. The efficacy of a pre-set ES protocol was evaluated by measuring the oxy-inflammation status in blood and urine and by questionnaire scales for fatigue and recovery assessments. As the authors conclude, the main message of this study is that there were not significant effects elicited by ES in treating DOMS, with exception of a subjective positive role, derived from VAS scores for muscle pain assessment.

The manuscript is interesting, well written and fluent, and scientifically sounds.

The authors greatly thank the reviewer for his/her positive opinions expressed on their research and on the manuscript style and sound.

The manuscript was checked/revised by Mr Grahame Humphrey, native English speaker who has a TEFL (Teaching English as Foreign Language) certificate and taught English for several years.

 General comments:

Line 189-190: The authors say: The device stimulated at a 0-100 Hz range frequency, by a 500μA complex waveform, auto-determined by the instrument without eliciting any muscular contraction. The authors should provide a more detailed description of the ES protocol, indicating the pulse duration, and the frequency applied. Also is not clear if the stimulation was a bilateral stimulation, and voltage applied.

  1. The authors thank the reviewer for the remarks. The raised points are clarified below and in the revised manuscript (see Materials and Methods).

Electro-stimulation devices are adopted for both Acupuncture and Transcutaneous electrical nerve stimulation (TENS).  The devices deliver electrical impulses through electrodes placed on the skin surface near nerves or at trigger points. Classical Acupuncture points, named meridians, are proximate to peripheral nerves. Given its wide diffusion, many electrical devices have been proposed to stimulate these points. Electrical devices suitable for Acupuncture are described in US 3,900,020, US 7,076,293 ed US 6,301,500 American Patents. One of the most important points to establish treatment effectiveness is described in US 3,900,020 Patent where the instrument is designed to generate electric signals different in frequency and waveform to be delivered to the electrodes.

In a similar way, the stimulation device adopted in our studies is equipped with: a signal generator, adapted to generate an electrical signal; an electrode, electrically connected to the signal generator itself, to receive the electrical signal generated by the signal generator; a control unit, operatively connected to the signal generator to adjust the parameters of the generated electric signal and to read the parameters of the electric signal returning from the electrode.

In other words, the electrical signal delivered to the tissue is in some way adapted to the tissue itself: it is a function of the measured tissue impedance, that in turn depends on the electric potential. The algorithm can deliver to the portion of tissue, where the electrode is placed, a constant electric potential in the 0-30 mV range. The instrument operation is automated and sequential, considering voltage and current levels as well as waveform, frequency, phase, and duration of the electric signal.

Figure 1.The stimulation signal stx is converted in a digital connector signal sc, delivered by the control unit. As shown in the Figure, sc signal is obtained by combining stx and a returning signal sr generated by the reflection of stx that is calculated by the difference between the impedance of the electrode and the impedance of the tissue where the electrode is applied.

Technical Specifications of the Instrument

Security Class: IIA

Nominal Tension: 110-230V/50-60 Hz

Absorbed Current: 150mA

Absorbed Power: 25W

Delivered Current: up to a 600 μA

Frequency: 0-100 Hz by complex waveforms

The stimulation adopted in our study was as follows: monolateral in the laboratory study (1); bilateral in the field study (2)

The second study was a high altitude ultra-endurance race, would be interesting include in discussion a consideration on possible differences observed across the two studies in view of a possible impact of environment hypoxia on the analysed biomarkers and parameters.

The authors thank the reviewer for his interesting remark. Indeed, the second study suffers the impact of environment hypoxia on the analyzed biomarkers and parameters, overall ROS levels and all parameters related to the metabolic and oxidative responses. This was the main reason why the results obtained by the two studies were not compared with each other. The effect of hypoxia just on ultra-endurance racers were previously studied by some of us (Schenk K, et al. Front Physiol. 2021 Nov 18;12:764694. doi: 10.3389/fphys.2021.764694 Mrakic-Sposta S, et al.. PLoS One. 2015 Nov 5;10(11):e0141780. doi: 10.1371/journal.pone.0141780). These considerations and the reference were reported in the Discussion section.

Reviewer 2 Report (New Reviewer)

The authors present a study aimed to investigate the efficacy of a specific electrical stimulation (ES) treatment on two different scenarios. The first scenario involved eleven healthy volunteers who underwent upper limbs exercise in a laboratory setting. The second scenario involved fourteen ultra-endurance athletes who participated in an ultra-running race. The paper is well written and structured but, in the reviewer’s opinion, a more detailed discussion of the obtained data should be included. Part of the discussion section includes information that should be moved to the introduction and results sections. 

Some specific comments

Abstract. 

Lines 27-29 Is this initial idea of comparing long exercise with moderate/vigorous exercise developed or discussed later in the text? Is it worthy to be mentioned? Please explain.

Table 3. It seems like equal signs are missing here. Same in Table 5?

Fig. 5. The lower limits of the legends are not the same for all the images. This makes difficult to compare the contour plots.

Table 4. I’d include here the time it took for the athletes to finish the race.

The quality of the language is correct.

Author Response

Comments and Suggestions for Authors

The authors present a study aimed to investigate the efficacy of a specific electrical stimulation (ES) treatment on two different scenarios. The first scenario involved eleven healthy volunteers who underwent upper limbs exercise in a laboratory setting. The second scenario involved fourteen ultra-endurance athletes who participated in an ultra-running race. The paper is well written and structured but, in the reviewer’s opinion, a more detailed discussion of the obtained data should be included. Part of the discussion section includes information that should be moved to the introduction and results sections. 

The authors greatly thank the reviewer for his/her positive opinions expressed on their research and on the manuscript style and sound and appreciated the helpful comments.

More detailed points were added in the Discussion Section. Two references were added.

As suggested, the sentence at the beginning of the Discussion was moved to the Introduction.

As concerns the moving from Discussion to Results, the authors want to highlight that, in submitting the study, they choose to present the data in separated Results and Discussion Sections. However, in the authors’ opinion, to perform a precise and detailed Discussion of the data, mostly for clarity and to help the reader, it is better to recall some data in the Discussion. On this basis, we didn’t move information from Discussion to Results.

Some specific comments

 Abstract. 

 Lines 27-29 Is this initial idea of comparing long exercise with moderate/vigorous exercise developed or discussed later in the text? Is it worthy to be mentioned? Please explain.

The authors completely agree with this reviewer’s remark. The comparison between long and moderate exercise is out from the aim of the study and as such isn’t worthy to be mentioned. Lines 27-29 were eliminated, and the text adjusted accordingly.

Table 3. It seems like equal signs are missing here. Same in Table 5?

The authors thank the reviewer: missing equal signs were added in both Table 3 and 5.

Fig. 5. The lower limits of the legends are not the same for all the images. This makes difficult to compare the contour plots.

The authors agree with the reviewer: the sentence ‘all images are displayed at the same thermal scale’ (line 361 and Caption of Figure 5) is misleading. As a matter of facts, to be able to compare images and calculate skin temperature data, all images were processed in Absolute Intensity mode, that is all values were scaled with respect to the maximum reference level. At the same time, the same color-coded range was used. These two assumptions implied that the lower limits didn’t result the same. The sentence was modified as: ‘all images were normalized to the maximum reference level’. in the text and in the Caption.

Table 4. I’d include here the time it took for the athletes to finish the race.

  1. The authors thank the reviewer for the suggestion. The race completion time was added in Table 4.

Comments on the Quality of English Language

The quality of the language is correct.

The manuscript was checked/revised by Mr Grahame Humphrey, native English speaker who has a TEFL (Teaching English as Foreign Language) certificate and taught English for several years.

Reviewer 3 Report (New Reviewer)

The authors state that ES does not treat DOMS as well as does not help in promoting muscle recovery, however they show in results significant differences between sham and ES in time course of blood lactate concentration; interleukin 6; VAS score; skin temperature.

This discrepancy should be discussed and also the significances should be stressed out better in results.  

Flaws in wording: line 97, 98: ...these latter by lower limbs

...in puctuation: line 117: ...,too

In general the text should be revised by a native speaker.

Author Response

Comments and Suggestions for Authors

The authors state that ES does not treat DOMS as well as does not help in promoting muscle recovery, however they show in results significant differences between sham and ES in time course of blood lactate concentration; interleukin 6; VAS score; skin temperature.

This discrepancy should be discussed and also the significances should be stressed out better in results. 

The authors thank the reviewer for the remark. To better clarify this point, the following statements were added to the Discussion.

‘The increase/decrease of the assessed biomarkers reported in Fig. 3 at T1 were the same for both experimental sessions and they were due to the performed exercise. After the ES (both Sham and Active) no significant differences were observed between the two treatments. Only a significant difference in VAS score after 1h, persisting at 48h post ES between Sham and Active groups was observed.

Similarly, the variations (T0 vs T1) reported in Fig. 6, 7 and 8 were ascribable to the race. Between the values recorded at T1 after the ES (both Sham and Active) no significant differences were observed. Again, in Fig. 8, no differences between the values recorded at T1 (post ES, Sham or Active) and at T2 (1 hour after ES, Sham or Active) were observed.’

Comments on the Quality of English Language

Flaws in wording: line 97, 98: ...these latter by lower limbs

...in puctuation: line 117: ...,too

In general the text should be revised by a native speaker.

  1. The authors apologize for the errors, now corrected. The manuscript was generally revised by a native speaker.

The manuscript was checked/revised by Mr Grahame Humphrey, native English speaker who has a TEFL (Teaching English as Foreign Language) certificate and taught English for several years.

Round 2

Reviewer 2 Report (New Reviewer)

Authors have considered my comments and suggestions in this version of the manuscript

Reviewer 3 Report (New Reviewer)

The paper now has significantly improved. Questions were clarified.

This manuscript is a resubmission of an earlier submission. The following is a list of the peer review reports and author responses from that submission.

Round 1

Reviewer 1 Report

Dear authors,

Thank you for your work and for submitting the manuscript to peer review. Please find below my comments.

General comments:

Please revise the texts throughout the document. There are many instances where sentences seem a little broken, as when a sentence was re-written but not harmonized with the surrounding text (e.g. lack of “and”, “the” or commas or other connecting word).

While the rationale of the study and its objectives are well presented, it is not clear the type of fatigue inducing exercise. From the abstract and introduction, the reader may infer it is related to running (hence, the ultra-runners). However, at the middle to end of the methods, it starts to get obvious it had something to do with the upper limbs (hand grip). Then, the authors state that not only does the study deal with two experimental environments (laboratory and field), there is also two experimental conditions (upper limbs and lower limbs).

During results, the authors report statistical differences as being p>0.05, p<0.01 and p<0.0001. I believe this last one has an additional zero, and that the authors intention as to state p<0.001.

Despite having an a priori power calculation, it would have been interesting o had the observed power to the statistical results.

Specific comments:

Line 63-71: The fourth paragraph of the introduction would be more appropriate after the fifth (line 72-76) or sixth (77-84) paragraph. The paragraph states the most effective recovery methods, but the fifth paragraph is still presenting why DOMS is an issue (to athletes) and the sixth paragraph which are some of the available therapeutics.

Line 117-121: I would argue that the analyzed parameters here described are not anthropometric in nature, but only physiological (as denoted by the section title). Body composition, blood pressure, and heart rate are physiological, but there is no description of the body shapes and sizes (anthropometry) such as mass, height, or girths. I would suggest revising the section title.

Line 129-130: the text says “each subject was requested to locate the felt pain”. Does this mean that the participants had to locate the position on their body where the pain was felt, or they had to mark on the VAS the subjective amount of (overall) pain they were feeling? Was there only a VAS scale for all-body pain sensation or, for instance, one for each lower limb?

Line 134-138: the microvette used to sample blood are reported to being produced by Sarstedt. However, plasma were obtained by centrifugation with a device that is not described.

Line 157: it is not clear is the blood used for lactate analysis was retrieved from the same batch of 500 ul blood sample (line 134), or if a new sample was drawn from a fingertip (as reported here).

Line 182-183: the authors refer an “exercised arm”. However, earlier in the manuscript it was stated that the exercise being tested was running. How does the upper limbs relate to running?

Line 186-189: this section is confusing. The participant performed a handgrip test with one of the upper limbs (the exercised one, according to line 1839. Then, why only the dominant upper limb was kept 90º flexed? And what happened when the dominant limb and the exercised limb were/weren’t the same? Do the authors meant to say that this was the testing position?

Line 197: is the “biceps” reported as “condition A” the biceps brachii (upper arm) or the biceps femoris (thigh)?

Line 199: if the device reported as used in line 194 was capable of delivering 500 uA of current at a frequency of 1-100 Hz, how did the authors managed to deliver 600 uA at 0-100 Hz?

Line 201: fake stimulation is something difficult to achieve, as most people will not feel the involuntary contractions of the muscle, nor fell the electric current. How did the authors manage this issue?

Line 212: according to the authors description the movement performed (biceps curls) was not purely eccentric, but rather a concentric (flexion) and eccentric (extension) movement.

Line 214: why did the authors selected a 3 kg mass for this exercise? Why not a mass corresponding to a given percentage of the participant’s own mass?

Line 215: it is not clear what “3 x s” means.

Line 212-221: there is no description of the position of the participant, other that the fact he was sitting. However, it would be interesting to understand if any support to the arm was provided, if the trunk was leaning to the front, If the movement started from full extension, and if the “full flexion” was considered at 90º of if higher flexion angles where allowed. The fact that the authors state that 90º was isometric is not decisive to these matters.

Line 222-225: each participant performed the “ES Sham” and “ES Active” with a 15 day interval, correct? Was this the effective order of treatment, or was it randomized?

Line 241: the “10,970 m+” reported is the summation of altitude differences? It should be reported as such, since it is no expectable that the absolute altitude differential (min. vs max. altitude) to be this. Also, each mountain height is reported without thousands separator, but this value presents a comma.

Line 267: the authors stated “normality was tested with the Shapiro Wilk’s test”. However, here they state that “the normality was tested by experimental data compared using ANOVA”.  Not only ANOVA is not a test of normality, but the normality test has already been reported.

Line 272-274: while the authors may have obtained a prospective sample result of 11 participants, meaning the recruited sample of 25 was (as reported by the authors) enough, this is not true to the previous laboratory study. That one had exactly 11 participants, the borderline condition.

Table 1. the unit of kilograms should not be reported with a capital K (Kg) but with a lower-case (kg).

Table 2. According to Section 2.1, physiological parameters included HR, SaO2 and blood pressure. However, in this table the blood pressure is not presented.

Line 295-296: This sentence does not seem to make sense. The authors state there was a significant increase in ROS at the end of the exercise (END) in the Sham and Active ES protocol, with values being presented. If the evaluation of the END condition was performed immediately after exercise, the fact there is no difference of ROS between basal and END conditions does not mean a “return toward the resting levels in the time course of the recovery”, because there was not time for recovery. That could be true if the authors were presenting the results after the 10 min of ES, which does not seem to be the case. On top of this, according to Figure 3A, the difference between REST and POST ES is (red #) p<0.001, which means that ROS values at POST ES did not return to REST levels. All these observations also affect the reporting of results for blood lactate and TAC.

Line 330-336: despite the current section being named “Score (RPE, VAS) values are presented for the isometric handgrip strength. This is a force measurement, and not a score, as made clear by the fact the authors have provided different sections during the methods description (2.2. scales. 2.6. handgrip).

Line 344-352: the results here reported are encouraging to the authors objective of showing potential differences on the inflammation process by using ES. However, it is not clear in this section, nor in the corresponding methods section, how temperature was measured. The thermographic camera presents the temperature of the all body in its field of view. How did the authors measured the surface temperature? Was it limited to the biceps region? Was it used a dot (to measure pixel value), a rectangle (to measure the average temperature), a poly line (to measure the average temperature)? Adding the analysis method (dot, rectangle, poly line) to the figure would be interesting to the reader understand the limits of temperature measurement.

Line 344-352: if electrodes were placed on the biceps brachii for the ES, how did the authors managed the local changes of temperature caused by the electrodes? Where they removed, bypassed, or not considered?

Table 2. Some variables do not have their unit presented (e.g. SaO2), other have a non-standard abbreviation (e.g. years = yr), or inconsistent unit description (e.g. Kg in one kg in another).

Line 378: it seems that the # meaning is missing a number (#p<0.0).

Line 400: “ns = p: not significant”. It would be better to present it as either “ns = not signicant” or “ns: p>0.05”.

Figure 6. These plots should be revised. In several instance (lets take panel A as example) we have a horizontal line stating there is no difference (ns between ES_Active T1 vs ES_Sham T0), and then we have other horizontal lines with no meaning (e.g. ES_Active T0 vs T1). Do these lines represent a difference? Represent a no difference as the other horizontal line? The figure legend state that * and ** represent differences, but these symbols are not present in Figure 6.

The language quality needs some improvements, especially in terms of grammar. There are several instances where a "and" or "the" is missing, when ";" is used instead of a dot, and placed where the sentence seems to have been re-written, but not re-read.

Author Response

We thank the reviewer for his/her constructive criticism: we responded item by item to all their raised questions, modifying the text accordingly. All changes made to the text of the revised version are reported in red. 

Dear authors,

Thank you for your work and for submitting the manuscript to peer review. Please find below my comments.

General comments:

Please revise the texts throughout the document. There are many instances where sentences seem a little broken, as when a sentence was re-written but not harmonized with the surrounding text (e.g. lack of “and”, “the” or commas or other connecting word).

While the rationale of the study and its objectives are well presented, it is not clear the type of fatigue inducing exercise. From the abstract and introduction, the reader may infer it is related to running (hence, the ultra-runners). However, at the middle to end of the methods, it starts to get obvious it had something to do with the upper limbs (hand grip). Then, the authors state that not only does the study deal with two experimental environments (laboratory and field), there is also two experimental conditions (upper limbs and lower limbs).

We thank the reviewer for appreciating our study and presentation and well agree with all his/her observations. The text was revised throughout the manuscript and harmonized with the surrounding. All changes are reported in red.“Upper limbs” in laboratory test and “lower limbs” in field experiment indications have been added in the Abstract, Introduction (see aim paragraph) and at the beginning of Methods sections.

During results, the authors report statistical differences as being p>0.05, p<0.01 and p<0.0001. I believe this last one has an additional zero, and that the authors intention as to state p<0.001.

We fully agree with the reviewer’s observation. Statistical differences were indicated as follows: *=0.05; **=0,01; ***=0,001; ****=,0001

Despite having an a priori power calculation, it would have been interesting o had the observed power to the statistical results.

The a priori calculation served us to check the number of subjects to test. When a posteriori calculated, by the obtained data (again using ROS data), at 80% power, we obtained a practically unchanged result.

Specific comments:

Line 63-71: The fourth paragraph of the introduction would be more appropriate after the fifth (line 72-76) or sixth (77-84) paragraph. The paragraph states the most effective recovery methods, but the fifth paragraph is still presenting why DOMS is an issue (to athletes) and the sixth paragraph which are some of the available therapeutics.

Thanks, according to this reviewer’s suggestion, paragraphs 4 and 5 have been inverted.

Line 117-121: I would argue that the analyzed parameters here described are not anthropometric in nature, but only physiological (as denoted by the section title). Body composition, blood pressure, and heart rate are physiological, but there is no description of the body shapes and sizes (anthropometry) such as mass, height, or girths. I would suggest revising the section title.

We agree: the section title has been changed in “physiological parameters” as well as the first sentence of the text.

Line 129-130: the text says “each subject was requested to locate the felt pain”. Does this mean that the participants had to locate the position on their body where the pain was felt, or they had to mark on the VAS the subjective amount of (overall) pain they were feeling? Was there only a VAS scale for all-body pain sensation or, for instance, one for each lower limb?

The sentence has been changed in “Each subject was asked to locate her/his body position where the pain was felt”.

Line 134-138: the microvette used to sample blood are reported to being produced by Sarstedt. However, plasma were obtained by centrifugation with a device that is not described.

The reference to the centrifuge device used to separate plasma has been added to the text.

Line 157: it is not clear is the blood used for lactate analysis was retrieved from the same batch of 500 ul blood sample (line 134), or if a new sample was drawn from a fingertip (as reported here).

The sentence of this paragraph was modified as follows: Another capillary blood sample (0,2 μL) was obtained from a fingertip for the determination of blood lactate concentration ([La]b) (Lactate scout; EKF, Italia, Milano, Italy).

Line 182-183: the authors refer an “exercised arm”. However, earlier in the manuscript it was stated that the exercise being tested was running. How does the upper limbs relate to running?

We agree and thanks for the remark. This paragraph has been rewritten.

Line 186-189: this section is confusing. The participant performed a handgrip test with one of the upper limbs (the exercised one, according to line 1839. Then, why only the dominant upper limb was kept 90º flexed? And what happened when the dominant limb and the exercised limb were/weren’t the same? Do the authors meant to say that this was the testing position?

We agree. Before performing the test, each subject was asked to indicate the dominant upper limb: All resulted right-handed. Then each participant, in a sitting position, performed a complete flexo- estension, 3kg load. The text was modified accordingly.

Line 197: is the “biceps” reported as “condition A” the biceps brachii (upper arm) or the biceps femoris (thigh)?

We thank the reviewer for his/her request for clarification. Experimental condition A was referred to the laboratory study A and it was the biceps brachii (upper arm). In the experimental condition B (referred to as the field study B) the electrical stimulation was applied to quadriceps. The text was modified accordingly.

Line 199: if the device reported as used in line 194 was capable of delivering 500 uA of current at a frequency of 1-100 Hz, how did the authors managed to deliver 600 uA at 0-100 Hz?

Thanks, it was a typing mistake. We have corrected in 500 uA.

Line 201: fake stimulation is something difficult to achieve, as most people will not feel the involuntary contractions of the muscle, nor fell the electric current. How did the authors manage this issue?

The supplied electric current was not perceptible, so that there was not any recognizable difference between active or sham stimulations. This clarification was added to the text.

Line 212: according to the authors description the movement performed (biceps curls) was not purely eccentric, but rather a concentric (flexion) and eccentric (extension) movement.

We agree with this reviewer observation: concentric was added in the text.

Line 214: why did the authors selected a 3 kg mass for this exercise? Why not a mass corresponding to a given percentage of the participant’s own mass? 

We thank the reviewer for his/her remark. We admit that it was a crucial decision! To select the 3Kg weight for the exercise, we started from the work by Lee et al. (ref 10). Indeed, in his study, the author starts the exercise loading the subjects by a very high weight (150% MIF), then lowering it in the time course of the experiment up to more than 4Kg lower. We instead decided to select a weight which the subjects (amateurs, not professional athletes) would be able to sustain all over the test. Different weights (2-5Kg) were tested before the experiment and 3Kg resulted the most suitable to be adopted, responding to the general aim of standardizing the study.  

Line 215: it is not clear what “3 x s” means.

We agree and thank the reviewer. 3s was the time lasted form one flexo-extension exercise to the other. The text has been modified accordingly.

Line 212-221: there is no description of the position of the participant, other that the fact he was sitting. However, it would be interesting to understand if any support to the arm was provided, if the trunk was leaning to the front, If the movement started from full extension, and if the “full flexion” was considered at 90º of if higher flexion angles where allowed. The fact that the authors state that 90º was isometric is not decisive to these matters.

We completely agree with the reviewer’s remark. First, before performing the test, each subject was asked to indicate the dominant upper limb: All resulted right-handed. Then each participant, in a sitting position, performed a complete flexo- estension, 3kg load. The text was modified accordingly.

Line 222-225: each participant performed the “ES Sham” and “ES Active” with a 15 day interval, correct? Was this the effective order of treatment, or was it randomized?

We agree, and added the following sentence: The two conditions were applied randomly to the subjects that therefore didn’t know which condition were submitted to.

Line 241: the “10,970 m+” reported is the summation of altitude differences? It should be reported as such, since it is no expectable that the absolute altitude differential (min. vs max. altitude) to be this. Also, each mountain height is reported without thousands separator, but this value presents a comma.

Thanks for the remarks: all corrected.

Line 267: the authors stated “normality was tested with the Shapiro Wilk’s test”. However, here they state that “the normality was tested by experimental data compared using ANOVA”.  Not only ANOVA is not a test of normality, but the normality test has already been reported.

Thank for the remark. The sentence was corrected.

Line 272-274: while the authors may have obtained a prospective sample result of 11 participants, meaning the recruited sample of 25 was (as reported by the authors) enough, this is not true to the previous laboratory study. That one had exactly 11 participants, the borderline condition.

To clarify, we have changed the sentence as follows:At a power of 80%, the calculated number of significant subjects was 11, which was sufficient for the laboratory study and below the subject’s population number recruited for the field study (B).

Table 1. the unit of kilograms should not be reported with a capital K (Kg) but with a lower-case (kg).

Thanks, All corrected.

Table 2. According to Section 2.1, physiological parameters included HR, SaO2 and blood pressure. However, in this table the blood pressure is not presented.

We agree with this reviewer: the paragraph was not clear. Blood Pressure was measured only in the Field Study (B). In fact, the variable is not reported in Table 2 but is reported in Table 4. This point is now specifically reported in the text.  

Line 295-296: This sentence does not seem to make sense. The authors state there was a significant increase in ROS at the end of the exercise (END) in the Sham and Active ES protocol, with values being presented. If the evaluation of the END condition was performed immediately after exercise, the fact there is no difference of ROS between basal and END conditions does not mean a “return toward the resting levels in the time course of the recovery”, because there was not time for recovery. That could be true if the authors were presenting the results after the 10 min of ES, which does not seem to be the case. On top of this, according to Figure 3A, the difference between REST and POST ES is (red #) p<0.001, which means that ROS values at POST ES did not return to REST levels. All these observations also affect the reporting of results for blood lactate and TAC.

The sentence has been changed as follows:After the recovery time (10 min), after both ES Sham and Active treatments, the ROS levels were near the basal values but anyway showing a low significant difference (p<0.01).

Line 330-336: despite the current section being named “Score (RPE, VAS) values are presented for the isometric handgrip strength. This is a force measurement, and not a score, as made clear by the fact the authors have provided different sections during the methods description (2.2. scales. 2.6. handgrip).

We fully agree with the reviewer: Figure 4 report both Vas scores results (A, B) and handgrip strength (4C). Therefore, the Title of the paragraph was changed as: Isometric handgrip strength and Scores (RPE, VAS).

Line 344-352: the results here reported are encouraging to the authors objective of showing potential differences on the inflammation process by using ES. However, it is not clear in this section, nor in the corresponding methods section, how temperature was measured. The thermographic camera presents the temperature of the all body in its field of view. How did the authors measured the surface temperature? Was it limited to the biceps region? Was it used a dot (to measure pixel value), a rectangle (to measure the average temperature), a poly line (to measure the average temperature)? Adding the analysis method (dot, rectangle, poly line) to the figure would be interesting to the reader understand the limits of temperature measurement.

Thanks for the remarks. A sentence to clarify the method was added in the paragraph 2.5.

Line 344-352: if electrodes were placed on the biceps brachii for the ES, how did the authors managed the local changes of temperature caused by the electrodes? Where they removed, bypassed, or not considered?

Temperature data were calculated by the Software from the Regions of Interest (ROI) delimited on the acquired images.

Table 2. Some variables do not have their unit presented (e.g. SaO2), other have a non-standard abbreviation (e.g. years = yr), or inconsistent unit description (e.g. Kg in one kg in another).

All unit variables were added and abbreviations have been corrected.

Line 378: it seems that the # meaning is missing a number (#p<0.0).

It has been corrected. Statistically significant differences symbols: *p<0.05; §, ***p<0.001; ¶, ****p<0.0001.

Line 400: “ns = p: not significant”. It would be better to present it as either “ns = not signicant” or “ns: p>0.05”.

 “ns= not significant” was deleted, since all lines referred to ns have been deleted in the figure.

Figure 6. These plots should be revised. In several instance (lets take panel A as example) we have a horizontal line stating there is no difference (ns between ES_Active T1 vs ES_Sham T0), and then we have other horizontal lines with no meaning (e.g. ES_Active T0 vs T1). Do these lines represent a difference? Represent a no difference as the other horizontal line? The figure legend state that * and ** represent differences, but these symbols are not present in Figure 6.

As reported below, the lines corresponding to not significance have been deleted. In all the figures tables the following symbols were adopted: *p<0.05; **p<=0.01; ***p<0.001; ****p<0.0001

Comments on the Quality of English Language

The language quality needs some improvements, especially in terms of grammar. There are several instances where a "and" or "the" is missing, when ";" is used instead of a dot, and placed where the sentence seems to have been re-written, but not re-read.

The text was improved and English Language revised.

Reviewer 2 Report

DEAR AUTHORS:

This study aimed to investigate the efficacy of electrical stimula-tion treatment, in the laboratory after eccentric exercise and in field after an ultra-running race.

After the first review some Deep changes are proposed in order to improve the final versión.

ü  Line 42, “intracellular proteins, such as creatine kinase and 42 lactate dehydrogenase [4,5]”. BOTH ARE ENZYMES

ü  Line 61, 62,” these observations are considered acute markers of exercise-induced muscle 61 damage and can provide an indication for the subsequent recoveryhese observations are considered acute markers of exercise-induced muscle 61 damage and can provide an indication for the subsequent recovery”. IT NEEDS REFERENCE. HERE YOU HAVE PROPOSALS RELATED TO RECOVERY AND MUSCLE DAMAGE IN LONG DISTANCES

Exercise-Induced Muscle Damage and Cardiac Stress During a Marathon Could be Associated with Dietary Intake During the Week Before the Race.

Mielgo-Ayuso J, Calleja-González J, Refoyo I, León-Guereño P, Cordova A, Del Coso J.Nutrients. 2020 Jan 25;12(2):316. doi: 10.3390/nu12020316.

Cryotherapy Models and Timing-Sequence Recovery of Exercise-Induced Muscle Damage in Middle- and Long-Distance Runners.

Qu C, Wu Z, Xu M, Qin F, Dong Y, Wang Z, Zhao J.J Athl Train. 2020 Apr;55(4):329-335. doi: 10.4085/1062-6050-529-18. Epub 2020 Mar 11. PMID: 32160058 Free PMC article.

ü  Line 64, “blood lactate”, BETTER BLOOD LACTATE CONCENTRATION

ü  Line 82, “The potential benefits purported by these interventions are those of an altered hemodynamics in order to facilitate a greater removal of tissue damaging molecules and a reduction in localized oedema formationThe potential benefits purported by these interventions are those of an altered hemodynamics in order to facilitate a greater removal of tissue damaging molecules and a reduction in localized oedema formation”. INCLUDE REFERENCES

ü  Line 94, “The research has been conducted on two different scenarios: in the laboratory, on healthy volunteers after eccentric exercise, and in a field study on ultra-endurance athletes after an ultra-running race (Tot Dret). The method herein pre sented adopted reliable, simple, and micro-invasive measurements to test the ES efficacy by monitoring the oxy-inflammation status throughout the assessment of many different parameters among which: ROS production levels investigated by EPR technique, Total Antioxidant Capacity (TAC) by electrochemical analyzer (EDEL), IL-6 by ELISA, Creatinine and Neopterin concentrations by HPLC methods. THIS PART OUTSIDE OF THE INTRODUCTION IS INSIDE THE METHODS

ü  Line 102. Participants inside the material and methods

ü  Line 104, Participants better tan subjects. Correct troguht the text

ü  Line 103-106, Participants. Did you included and exclusion criteria. Besides no Ethical Issues are registered, for example Ethical Committe all procedures are done acoording to Helsinki Declaration. MOVE FROM LAST PART TO THE PARTICIPANTS PARAGRAPH

ü  Line 106. Did you include ethical Comitte number?

ü  Line 116, “2.1 Anthropometric and physiological parameters”. antropometrhist Include validity and reliability of devices and Register, and ISSAK registration of the

ü  Line 157, “2.3.3 Lactate. Capillary blood (0.2 μL) was obtained from a fingertip for the determination”. BETTER BLOOD LACTATE CONCENTRATION

ü  Line 262, Statistical analysis. Did you check the normality of the data before. And did you applied the LEVENE Test to check the Homocedasticity of the data? And POST HOC did you used. Which one?  

ü  Methods in general: Devices and company include ® in superindex

ü  Line 282. Table 1. Body mass better than weight, Correct trought the text. Move this table inside the participants part

ü  After participants, you have to include the general procedure

ü   

ü  Describes all performance parameters (Validity and reliability). Include as well previous studies that used these variables.

ü  Line 432, in the first paragraph of the disscussion include the main results.

ü  Line 462, “Initial nociceptors stimulation by products of tissue breakdown and oedema is thought to occur within the first day of damage. In the present study, on the basis of the thermographic images, an ES treatment effect at this early phase through an analgesic effect of stimulation of afferent nerve fibres as well as a reduction in oedema brought by the increased microvascular blood flow could be hypothesized. Initial nociceptors stimulation by products of tissue breakdown and oedema is thought to occur within the first day of damage. In the present study, on the basis of the thermographic images, an ES treatment effect at this early phase through an analgesic effect of stimulation of afferent nerve fibres as well as a reduction in oedema brought by the increased microvascular blood flow could be hypothesized”. INCLUDE REFERENCES

ü   

ü  Line 471, The only effect here found was a best lowering of the temperature, measured with thermography in the A post-exercise experimental condition, after Active ES with respect to Sham ES. The only effect here found was a best lowering of the temperature, measured with thermography in the A post-exercise experimental condition, after Active ES with respect to Sham ES. INCLUDE REFERENCES

ü  Line 496, Include practical applications and strenghts of your study and future research in future studies utilizing a similar protocol) in separate paragraph.

ü  Tables title up, no down

Quality of English must be done

Author Response

We thank the reviewer for his/her constructive criticism: we responded item by item to all their raised questions, modifying the text accordingly. All changes made to the text of the revised version are reported in red. 

Line 42, “intracellular proteins, such as creatine kinase and 42 lactate dehydrogenase [4,5]”. BOTH ARE ENZYMES

We thank this reviewer for his/her overall constructive criticism. 

 We agree: “intracellular proteins” was changed in “intracellular enzymes”.

Line 61, 62,” these observations are considered acute markers of exercise-induced muscle damage and can provide an indication for the subsequent recovery. These observations are considered acute markers of exercise-induced muscle 61 damage and can provide an indication for the subsequent recovery”. IT NEEDS REFERENCE. HERE YOU HAVE PROPOSALS RELATED TO RECOVERY AND MUSCLE DAMAGE IN LONG DISTANCES

We agree with this reviewer remark. The following references have been added (18 and 19) : Exercise-Induced Muscle Damage and Cardiac Stress During a Marathon Could be Associated with Dietary Intake During the Week Before the Race. Mielgo-Ayuso J, Calleja-González J, Refoyo I, León-Guereño P, Cordova A, Del Coso J.Nutrients. 2020 Jan 25;12(2):316. doi: 10.3390/nu12020316.

Cryotherapy Models and Timing-Sequence Recovery of Exercise-Induced Muscle Damage in Middle- and Long-Distance Runners.Qu C, Wu Z, Xu M, Qin F, Dong Y, Wang Z, Zhao J.J Athl Train. 2020 Apr;55(4):329-335. doi: 10.4085/1062-6050-529-18. Epub 2020 Mar 11. PMID: 32160058 Free PMC article.

Line 64, “blood lactate”, BETTER BLOOD LACTATE CONCENTRATION

We agree, the change was made in the text.

Line 82, “The potential benefits purported by these interventions are those of an altered hemodynamics in order to facilitate a greater removal of tissue damaging molecules and a reduction in localized oedema formation”.

Thanks, done. Ref number 34 was added.

Line 94, “The research has been conducted on two different scenarios: in the laboratory, on healthy volunteers after eccentric exercise, and in a field study on ultra-endurance athletes after an ultra-running race (Tot Dret). The method herein presented adopted reliable, simple, and micro-invasive measurements to test the ES efficacy by monitoring the oxy-inflammation status throughout the assessment of many different parameters among which: ROS production levels investigated by EPR technique, Total Antioxidant Capacity (TAC) by electrochemical analyzer (EDEL), IL-6 by ELISA, Creatinine and Neopterin concentrations by HPLC methods. THIS PART OUTSIDE OF THE INTRODUCTION IS INSIDE THE METHODS

Thanks, We have deleted the sentence, and inserted this part in the methods.

Line 102. Participants inside the material and methods

Indeed, participants have been inserted in the experimental protocols because they are different in the two parts of the study (laboratory and field studies).

Line 104, Participants better tan subjects. Correct troguht the text

We agree: subjects was changed in participants.

Line 103-106, Participants. Did you included and exclusion criteria. Besides no Ethical Issues are registered, for example Ethical Committe all procedures are done acoording to Helsinki Declaration. MOVE FROM LAST PART TO THE PARTICIPANTS PARAGRAPH.

In the paragraph 2 Materials and Methods, we reported that all procedures were conducted according to the Declaration of Helsinki and approval was obtained from the institutional Ethics Committee of the Aosta Hospital (n.895; 108 31/8/2015), Italy.

Line 106. Did you include ethical Comitte number?

Yes, we did.

Line 116, “2.1 Anthropometric and physiological parameters”. antropometrhist Include validity and reliability of devices and Register, and ISSAK registration of the

As suggested by the other reviewer, the title paragraph has been changed in “physiological parameters” because the parameters described were not anthropometric, but only physiological.

Line 157, “2.3.3 Lactate. Capillary blood (0.2 μL) was obtained from a fingertip for the determination”. BETTER BLOOD LACTATE CONCENTRATION

We have changed the title paragraph in “blood lactate concentration”

Line 262, Statistical analysis. Did you check the normality of the data before. And did you applied the LEVENE Test to check the Homocedasticity of the data? And POST HOC did you used. Which one? 

 Thanks for the remarks. Ttables 3 and 5 with LEVENE test results  have been added to the submitted revised version of the manuscript. Moreover, the experimental data have been compared using ANOVA repeated measures with a Dunn’s multiple comparison post-hoc test.

Methods in general: Devices and company include ® in superindex

Thanks, we have included the trademark symbols to devices and company

Line 282. Table 1. Body mass better than weight, Correct trought the text. Move this table inside the participants part

We thought they were synonyms. However, we thank and we have corrected throughout the text.

After participants, you have to include the general procedure

The participants are included into the specific procedure as the participants of experiment A were not the same of experiment B so we think that is more simply for reader.

Describes all performance parameters (Validity and reliability). Include as well previous studies that used these variables.

No specific performance parameters have been assessed and the only one present in the study is  the maximal isometric handgrip strength (N) that is a variable well known and validated by the international literature.

Line 432, in the first paragraph of the discussion include the main results.

A sentence has been added.“The data of the present study showed up that the ES treatment did not modify the values of the measured biological parameters. Indeed, the only effect here found was a best lowering of the temperature, measured with thermography in the A post-exercise experimental condition, after Active ES with respect to Sham ES and a positive perceptual benefit on fatigue sensation of the subjects.”

Line 462, “Initial nociceptors stimulation by products of tissue breakdown and oedema is thought to occur within the first day of damage. In the present study, on the basis of the thermographic images, an ES treatment effect at this early phase through an analgesic effect of stimulation of afferent nerve fibers as well as a reduction in oedema brought by the increased microvascular blood flow could be hypothesized.”.

A reference has been added (34):

Burgess, L.; Immins, T.; Swain, I.; Wainwright, T. Effectiveness of Neuromuscular Electrical Stimulation for Reducing Oedema: A Systematic Review. J Rehabil Med 2019, 51, 237–243, doi:10.2340/16501977-2529.

Line 471, The only effect here found was a best lowering of the temperature, measured with thermography in the A post-exercise experimental condition, after Active ES with respect to Sham ES.

This is a result of the present study.

Line 496, Include practical applications and strengths of your study and future research in future studies utilizing a similar protocol in separate paragraph.

Thanks, we added the separate paragraph: 5. Pratical Application and strengths of the study

Tables title up, no down

Done, thanks

Round 2

Reviewer 2 Report

Accepted

Accepted